# Endogenously Produced Jasmonates Affect Leaf Growth and Improve Osmotic Stress Tolerance in Emmer Wheat

**DOI:** 10.3390/biom13121775

**Published:** 2023-12-12

**Authors:** Alexey V. Pigolev, Dmitry N. Miroshnichenko, Sergey V. Dolgov, Valeria V. Alekseeva, Alexander S. Pushin, Vlada I. Degtyaryova, Anna Klementyeva, Daria Gorbach, Tatiana Leonova, Aditi Basnet, Andrej A. Frolov, Tatyana V. Savchenko

**Affiliations:** 1Institute of Basic Biological Problems, Pushchino Scientific Center for Biological Research, Russian Academy of Sciences, 142290 Pushchino, Russia; alexey-pigolev@rambler.ru (A.V.P.); miroshnichenko@bibch.ru (D.N.M.); 2Branch of Shemyakin and Ovchinnikov Institute of Bioorganic Chemistry, Russian Academy of Sciences, 142290 Pushchino, Russia; dolgov@bibch.ru (S.V.D.); lera@bibch.ru (V.V.A.); aspushin@gmail.com (A.S.P.); vlada.buryak@gmail.com (V.I.D.); anutik.vlasowa@yandex.ru (A.K.); 3Department of Bioorganic Chemistry, Leibniz Institute of Plant Biochemistry, 06120 Halle (Saale), Germany; daria.gorba4@yandex.ru (D.G.); tatiana.leonova@ipb-halle.de (T.L.); andrej.frolov@ipb-halle.de (A.A.F.); 4Laboratory of Analytical Biochemistry and Biotechnology, Timiryazev Institute of Plant Physiology, Russian Academy of Sciences, 127276 Moscow, Russia

**Keywords:** jasmonates, allene oxide synthase, 12-oxophytodienoate reductase, osmotic stress, tolerance, emmer wheat

## Abstract

In light of recent climate change, with its rising temperatures and precipitation changes, we are facing the need to increase the valuable crop’s tolerance against unfavorable environmental conditions. Emmer wheat is a cereal crop with high nutritional value. We investigated the possibility of improving the stress tolerance of emmer wheat by activating the synthesis of the stress hormone jasmonate by overexpressing two genes of the jasmonate biosynthetic pathway from *Arabidopsis thaliana*, *ALLENE OXIDE SYNTHASE* (*AtAOS*) and *OXOPHYTODIENOATE REDUCTASE 3* (*AtOPR3*). Analyses of jasmonates in intact and mechanically wounded leaves of non-transgenic and transgenic plants showed that the overexpression of each of the two genes resulted in increased wounding-induced levels of jasmonic acid and jasmonate-isoleucine. Against all expectations, the overexpression of *AtAOS*, encoding a chloroplast-localized enzyme, does not lead to an increased level of the chloroplast-formed 12-oxo-phytodienoic acid (OPDA), suggesting an effective conversion of OPDA to downstream products in wounded emmer wheat leaves. Transgenic plants overexpressing *AtAOS* or *AtOPR3* with increased jasmonate levels show a similar phenotype, manifested by shortening of the first and second leaves and elongation of the fourth leaf, as well as increased tolerance to osmotic stress induced by the presence of the polyethylene glycol (PEG) 6000.

## 1. Introduction

Among the monocotyledonous plants, wheat deserves special attention due to its practical importance and the significant gaps in basic knowledge of this crop. Tetraploid wheat varieties in particular have been poorly studied. This is despite their considerable practical potential. Emmer wheat (*Triticum dicoccum*; Schrank., 2n = 4x = 28; syn *Triticum turgidum L. subsp. dicoccum* [*Schrank ex Schubl.*] *Thell*) is considered to be the direct progenitor of modern cultivated wheat varieties, including tetraploid durum wheat (*T. durum* Desf., 2n = 4x = 28) and hexaploid bread wheat (*T. aestivum* L., 2n = 6x = 42). Compared to all wheat varieties and other cereals, the grain of the tetraploid wheat is characterized by a higher protein content, a low-calorie value, and a rich mineral composition, as well as the presence of vitamins [1,2]. All of the essential nutrients from the grain are easily digestible due to their high solubility in water. Emmer wheat is currently cultivated in many countries. The wild emmer wheat genome has been fully sequenced [3], which provides a basis for further research and a platform for the wider use of this valuable crop.

Current climate change has led to temperature rising and changes in rainfall. All of these have negative impacts on crops, which tend to be extremely susceptible to environmental stressors—drought, salinity, or unfavorable temperature conditions. More than that, world agriculture is now facing water shortages which exceed previous pessimistic projections [4,5,6]. Under these circumstances, the challenge of feeding the ever-expanding global population is becoming more and more serious. Therefore, any instrumentation that can help us in achieving yields improvement under adverse environmental conditions is valuable. One approach to enhance the plant stress tolerance is to alter the activity of signaling pathways associated with stress hormones. Among the major stress hormones some of the most important are fatty-acid-derived jasmonates [7,8,9].

ALLENE OXIDE SYNTHASE (AOS) represents the first enzyme in the jasmonate biosynthetic pathway. In chloroplasts, AOS converts 13-hydroperoxy-9,11,15-octadecatrienoic acid to 12-oxo-phytodienoic acid (OPDA) with the help of ALLENE OXIDE CYCLASE [10,11]. Peroxisome-localized 12-OPDA REDUCTASE (OPR) is an enzyme responsible for the reduction of OPDA [12]. As a result of OPR activity and three cycles of β-oxidation, jasmonic acid (JA) is formed [13]. Importantly, the OPDA is not only a primary precursor of JA but also an integral element in a variety of signaling pathways by itself [14]. The content of two key metabolites in the jasmonate biosynthesis pathway, OPDA and JA, can be regulated by altering the activity of AOS and OPR enzymes. This could lead to improved plant tolerance to adverse environmental conditions, including abiotic stresses [15].

Jasmonates (JAs) are involved in many crucial processes during adaptation to abiotic stress conditions such as regulation of stress-related gene expression, biosynthesis of protective compounds, and metabolism rewiring [7,16]. The protective effect of jasmonates under water-deficiency stress has been demonstrated in many plants, including *Arabidopsis thaliana* and numerous agricultural dicotyledonous and monocotyledonous species [17,18,19,20]. It has been shown that the protective functions of JA and OPDA may differ [21]. A large corpus of works demonstrates the protective effect of exogenous treatment using jasmonates. Thus, induction of drought tolerance through the application of exogenous methyl jasmonate has been previously reported for cauliflower [22], rice [23,24], and soybean [25]. The protective effect of jasmonates under drought/osmotic stress/salt-stress conditions is frequently associated with the regulation of the antioxidant system [9,22,26,27]. Crosstalk with other hormones, such as abscisic acid or cytokinin, is another possible mechanism underlying the protective functions of jasmonates [8,20,28]. Another possible effect is the protection of photosynthetic functions and biosynthesis of protective metabolites such as proline [8,29,30,31,32]. Regulation of stomatal closure as a key defense response during the course of water deficiency using JAs has also been reported [21,33]. To explore the possibility of enhancing plant-stress tolerance via genetic modifications and to elucidate the functions of individual jasmonates, we have generated transgenic emmer wheat plants overexpressing *AOS* or *OPR*.

## 2. Materials and Methods

### 2.1. Generation of Transgenic Emmer Wheat Plants Overexpressing AtAOS

The *Arabidopsis thaliana AtAOS* gene (AT5G42650, UniProt KB Q96242) does not contain introns; therefore, we obtained the full-length sequence of the *AtAOS* gene including the sequence for chloroplast targeting signal from Arabidopsis genomic DNA using PCR. Total DNA was extracted from wild-type leaves of the Columbia-0 ecotype of *Arabidopsis thaliana*, and amplification was performed using gene-specific primers: forward 5′-gatc**CCCGGG**TAGATGGCTTCTATTTCAACCCCT-3′ and reverse 5′-gatc**GAGCTC**TCCTAAAAGCTAGCTTTCCTTAAC-3, where SmaI (forward primer) and SacI (reverse primer) restriction sites are shown in bold and start and stop codons are underlined. After amplification, the produced PCR fragment (1582 bp) was purified via gel electrophoresis, treated with SmaI and SacI restriction enzymes, and ligated at the SmaI/~/SacI sites into the pUC19::PUbi1-Tnos vector between the Ubi1 promoter from maize and the Tnos terminator from *Agrobacterium tumefaciens*. Then, the fragment with the sequence of the target gene with regulatory sequences was cut out using PvuII, and the PUbi-AtAOS-Tnos fragment was transferred into the SmaI site of the psGFP-BAR vector [34], containing two selective marker genes: *GFP*, coding for green fluorescent protein, and *BAR*, coding for phosphinothricin acetyl transferase and conferring herbicide resistance (Figure 1). We have previously shown that wheat transformation with the psGFP-BAR vector does not result in any phenotypical alterations [35]. The molecular constructs were checked using the sequencing to ensure fidelity and integrity of DNA sequences.

The generated vector was introduced into emmer wheat (cv. Runo) using microprojectile bombardment of 10–15-day-old, scutellum-derived morphogenic calli as previously described [35]. Individual transgenic plants were selected based on their resistance to the herbicide phosphinothricin and the GFP fluorescence of cells and tissues. Transgenic plants were screened via PCR using the following primers: 5′-GCGACGTAAACGGCCACAAG-3′ (forward) and 5′-CCAGCAGGACCATGTGTGATCG-3′ (reverse) for the *GFP*; 5′-GAATCCGCCGGTGAGATTCTCGTTGAAGCCG-3′ (forward) and 5′-ACGAGAAATTAACGGAGCTTCCTAACGGCGACG-3′ (reverse) for the *AtAOS*. The same primers were also used to analyze the transgene expression in leaves of the primary T_0_ transgenic plants via end-point RT-PCR. The homozygous status of transgenic T_1_ sublines derived from self-pollinated T_0_ plants was determined using the GFP fluorescence of pollen, as previously described [36]. Sets of these sublines were chosen to generate T_4_ seeds for further analysis. Prior analysis the stable inheritance and expression of transgenes (*GFP* and *AtAOS*) in T4 progenies was confirmed using end-point RT-PCR.

### 2.2. Quantitative Real-Time PCR

For quantitative real-time RT-PCR, total RNA was extracted from the 4th young leaf of non-transgenic and transgenic emmer wheat with the Trizol reagent (Thermo Fisher Scientific, Waltham, MA, USA) according to the manufacturer’s protocol and used to synthesize cDNA as previously described [37]. According to the sequence of *AtAOS*, the primers were designed: 5′-AAATCCAACGGCGGAGAACT-3′ as forward primer and 5′-TCGTCGCCAACGGTTGATAA-3′ as reverse primer. Transcript levels were quantified using QuantStudio™ 5 Real-Time PCR Cycler (Thermo Fisher Scientific). The specificity of PCR amplification was confirmed by the presence of a single peak in the melting curve of products. The relative levels of *AtAOS* overexpression were calculated using the wheat *TaWIN1* gene as a reference gene, the primers for *TaWIN1* were as follows: 5′-TTTTCTGTGTTCTACTATGAGATCTTGAA-3′ as forward primer and 5′-AAGTGCATAATTAAACAGAGGTAGTGATG-3′ as reverse primer. As the parent emmer wheat does not express *AtAOS*, the lowest accumulation of *AtAOS* transcript found using qRT-PCR in one of the transgenic samples was taken as “1” to quantify the *AtAOS* expression levels in transgenic lines.

### 2.3. Phytohormone Analysis

The analysis of phytohormones in the third leaves of transgenic and non-transgenic emmer wheat at the four-leaf stage was carried out exactly as previously described [37]. All wounding experiments were accomplished in six independent repetitions, i.e., the plants were randomized and wounded at different times (with fixed time interval between wounding and harvest). For wounding, the third leaf was clamped with tweezers across the leaf blade along its entire length at 10 cm intervals. Leaf tissues were collected 30 min after wounding. Non-wounded control leaves were collected prior to the onset of wounding. After collection, the material was immediately ground in liquid nitrogen using a mortar and pestle, followed by grinding in a Mixer Mill MM 400 ball mill with 3 mm diameter stainless steel balls (Retsch, Haan, Germany) at a vibration frequency of 30 Hz for 2 min. Approximately 50 mg of the frozen ground plant material was extracted using 500 μL methanol containing 0.1 ng/μL of ^2^H_6_-JA, ^2^H_2_-(−)-JA-Ile and ^2^H_5_-OPDA. The procedure relied on 1.6 mL cryo-tubes (Precellys Steel Kit 2.8 mm, Peqlab Biotechnologie GmbH, Erlangen, Germany) and a bead mill (FastPrep24 instrument, MP Biomedicals LLC, Santa Ana, CA, USA) with acceleration of 6.5 m/s^2^ for 30 s. The suspensions were centrifuged (20,000× *g*, 2 min, 0 °C), diluted with water to 5 mL, and subjected to solid-phase extraction (SPE) as described by Balcke et al. [38].

LC-MS/SM analysis relied on reversed-phase, ultra-high-performance liquid chromatography coupled with a triple quadrupole tandem mass spectrometry (RP-UHPLC-QqQ-MS/MS) in the multiple reaction monitoring (MRM) mode [38], accomplished using an ACQUITY H-Class UPLC ultrahigh performance liquid chromatography system (Waters GmbH, Eschborn, Germany) coupled with a QTRAP 6500 (AB Sciex, Darmstadt, Germany) triple quadrupole-linear ion trap instrument operating in negative MRM mode under the instrument settings described by Leonova and co-workers [39].

### 2.4. Measurements of the Leaf Length

Non-transgenic and transgenic plants (T_4_ homozygous progenies) of emmer wheat (cv. Runo) were grown in a controlled-environment greenhouse in one-liter pots (two plants per pot) filled with soil. Pots were randomized on shelves; 15 pots per m^2^. Plants were cultivated under a 16 h photoperiod at 25 ± 2 °C during the day and 20 ± 2 °C during the night with additional lighting to provide light intensity up to 200 µmol m^−2^s^−1^. The measurement of the length of the 1st, 2nd, 3rd, and 4th leaves was carried out when the plants developed an opened 6th leaf. Obtained data were subjected to the one-way analysis of variance (one-way ANOVA); means were compared with the control group (Runo) using Dunnett’s multiple comparisons test.

### 2.5. Osmotic Stress Tolerance Test

Seeds were surface-sterilized via 30 s incubation in 80% ethyl alcohol followed by 20 min of incubation in 4% sodium hypochlorite solution. Then, seeds were rinsed thoroughly in sterile water and placed in plastic Petri dishes (9 cm diameter) lined with two layers of filter paper (15–20 seeds per dish). Seeds of only one genotype were placed in each dish. An amount of 8 mL of sterile water or polyethylene glycol (PEG) solution was added to the Petri dishes. Water or PEG solution was added to Petri dishes if necessary, during prolonged incubation. The light/dark photoperiod was established as 12:12 h (10–35 μmol photons m^−2^s^−1^) with an incubation temperature of 20 ± 1 °C. For seedlings grown in water, the longest roots and coleoptiles length were measured on the 4th day after seed soaking and on the 8th day for seedlings grown in 20% PEG 6000. An amount of 25% PEG 6000 almost totally suppresses the root and coleoptiles elongation in all studied lines. For the stress treatment, seeds were incubated in 25% PEG for 21 days; then, 8 mL of water was added to the Petri dishes, i.e., the PEG concentration was reduced twice and the lengths of roots and shoots were measured after 4 days. Seeds with coleoptile or root lengths exceeding 1 mm were considered germinated. To calculate the percentage of germinated seeds, independent experiments were conducted (three experiments on seed germination in water, three experiments on seed germination in 20% PEG 6000, and two experiments on seed germination in 25% PEG 6000) with at least 14 seeds of each genotype in each experiment. Two experiments to measure the seedling growth rate were carried out for each of the germination conditions, in 20% PEG 6000 and 25% PEG 6000, with at least 14 seeds of each genotype in the experiment.

## 3. Results

### 3.1. Generation of Transgenic Emmer Wheat Plants Overexpressing AtAOS

To stimulate jasmonate biosynthesis in emmer wheat, we overexpressed the *Arabidopsis thaliana ALLENE OXIDE SYNTHASE* gene, which has been extensively studied and whose role in jasmonate biosynthesis has been experimentally confirmed [40]. The genetic transformation of emmer wheat Runo was performed using the biolistic method [35]. Transformation vector pBAR-GFP.UbiAOS, carrying *AtAOS* and selective genes, used in this study (Figure 1) was essentially similar to the vector we used previously to generate transgenic hexaploid and tetraploid emmer wheat overexpressing the *OPR* gene [37].

Ten independent transgenic events were generated using 576 young morphogenic calli derived from cultured immature embryos (Appendix A). Transgenic plants were confirmed via PCR amplification of specific fragments of *GFP* and *AtAOS* genes (Appendix A). Most of the selected GFP-positive transgenic plants, except the RA1, showed the integration of the *AtAOS* sequence. The overexpression of introduced genes was further confirmed using RT-PCR (Appendix A). The six primary transgenic plants were found to constitutively express *AtAOS* in leaves. Three independent transgenic events showed no accumulation of the *AtAOS* transcripts (Appendix A) despite the integration of the *AtAOS* sequence (Appendix A). T_1_ plants of three expressing events (RA3, RA4, and RA7) were segregated in a 3:1 ratio (transgenic to non-transgenic) and, therefore, were likely to contain a single transgenic insertion (χ^2^ tested at *p* < 0.05). The T_0_ RA9 plant segregated transgenes with a 15:1 ratio (χ^2^ = 0.28 < χ^2^_0.05_ = 3.84), indicating that two insertions had occurred at unlinked loci (Appendix A). T_2_ homozygous sublines of three independent transgenic events (RA3, RA4, and RA9) were used for further studying.

The successful inheritance and stable overexpression of *AtAOS* in subsequent progenies was further confirmed using quantitative, real-time RT-PCR (qRT-PCR) (Figure 2). The abundance of *AtAOS* transcripts was easily observed in T_4_ plants of three selected homozygous sublines while the RT-PCR product was not detected in non-transgenic emmer wheat. Despite the significant fluctuation in results among individual T_4_ plants, the *AtAOS* transcript level was overall higher in intact leaves of RA3 and RA4, while the RA9 line showed a 5–10 times lower average expression level (Figure 2).

### 3.2. Characterization of the Transgenic Emmer Wheat Plants

#### 3.2.1. Phytohormone Analysis

To better understand the functionality of individual *AOS* and *OPR* genes of the JAs biosynthesis pathway when overexpressed in emmer wheat, we performed a comparative analysis of newly established transgenic lines with *AtAOS* overexpression (RA3, RA4, and RA9) and previously generated transgenic emmer wheat plants with *AtOPR3* overexpression (RC12, RC26, and R29) [37]. First, we analyzed the content of the main jasmonates in leaves of transgenic and non-transgenic plants, including OPDA, JA, and JA-Ile conjugate—the most active form of jasmonate, which performs most of the signaling functions. Basal levels of these metabolites in intact tissues are usually very low, so we induced jasmonate accumulation via mechanical wounding of the leaves. The phytohormone content was analyzed 30 min after the wounding (Figure 3). As expected, the overexpression of *AtOPR3* (that encodes a peroxisome-localized enzyme) led to an increase in wounding-induced jasmonic acid levels in leaf tissues, as previously shown [37]. However, against expectations, in cases of *AtAOS* overexpression encoding a chloroplast-localized enzyme, no increase in levels of the chloroplast-formed OPDA was observed, while the level of JA in mechanically damaged plants was increased significantly, in comparison to non-transgenic plants. All transgenic *AtAOS* lines demonstrated a statistically significant increase in the wounding-induced JA level. The level of JA-Ile was significantly increased in two of the three transgenic lines (RA3 and RA4). Although no statistical significance was revealed for the RA9 line due to a scatter of values obtained for individual plants, the average and median values of JA-Ile are noticeably higher in RA9 in comparison with non-transgenic plants. Interestingly, a tendency for a decrease in the basal level of JA and JA-Ile in transgenic lines in comparison to non-transgenic plants was also detected. The median JA and JA-Ile values in five out of six transgenic lines are lower than in the non-transgenic control, the difference being statistically significant in two transgenic lines (RC29 in JA; RC26 in JA-Ile).

#### 3.2.2. Analysis of Plants’ Growth Phenotype

To investigate the effect of jasmonate biosynthesis gene overexpression on early vegetative growth of transgenic emmer wheat, we monitored early seedling development and plant growth in soil pots up to the six-leaf stage. In water-germinated seeds, no difference in germination percentage was revealed as all the seeds germinated. There was no difference in the coleoptile length in seedlings growing under these conditions, while in transgenic plants noticeably longer roots were observed (Figure 4). The median root length of non-transgenic Runo seedlings was 3.7 cm, whereas the median length for all transgenic lines was at least 4 cm. Four of the six transgenic lines had median root lengths of 5 cm or more, with the longest root of 5.75 cm in RA4 plants.

Then, we examined the vegetative growth characteristics of transgenic plants grown in soil under climate-controlled greenhouse conditions. We measured the length of four fully developed first leaves of two *AtAOS* transgenic lines (RA4 and RA9) and two *AtOPR3*-expressing transgenic lines (RC12 and RC29) together with the non-transgenic plants (Runo). This type of measurement is simple but reliable and easily quantifiable, thus allowing us to detect a significant alteration in the growth pattern of transgenic plants (Figure 5). At the early vegetative stage, the apparent reduction in the first leaf length was observed for all transgenic lines expressing either *AtAOS* or *AtOPR3*. The first leaves of RA4 and RA9 were on average 13% shorter, while the first leaves of RC12 and RC29 showed a decrease in length by 19% and 15%, respectively (Figure 5A). A similar trend was observed for the second leaf; the difference with non-transgenic plants was small but still statistically significant (Figure 5B). Compared to the mean length of 29.8 ± 0.36 cm in non-transgenic plants (Runo), the length of the second leaves of RC12 and RC2 9 (*AtOPR3* overexpression) was 27.8 ± 0.57 cm (7% reduction) and 26.9 ± 0.46 cm (10% reduction), correspondently. A significant decrease in length by 14% (25.5 ± 0.57 cm) was still observed in the RA9, while the length of the second leaf of RA4 was only 5% shorter on average (28.2 ± 0.47 cm).

No significant difference in the length of the third leaves was found (Figure 5C). The average length of the third leaves of RA4, RA9, RC12, and RC29 ranged from 28.4 to 30.8 cm, while the average length of the third leaf of non-transgenic plants—29.6 ± 0.47 cm. Interestingly, the fourth leaf of transgenic plants outgrew control plants: both *AtAOS*-overexpressing lines showed significant (*p* < 0.01) leaf elongation (Figure 5D). Two *AtOPR3*-overexpressing lines also showed an increase in the fourth leaf length, but we could only confirm the difference between RC12 and non-transgenic plants (Runo) (33.4 cm vs. 30.58 cm, *p* < 0.005), while the fourth leaf of RC29 was only slightly longer than in non-transgenic plants (in average 3%), with a non-significant difference (*p* = 0.5886).

### 3.3. Analysis of Osmotic Stress Tolerance

#### 3.3.1. Effect of Osmotic Stress on Seed Germination

To observe the seed germination under osmotic stress conditions, appropriate conditions were created using PEG 6000. The seed germination was evaluated by counting seedlings with clearly visible emerging roots (bigger than 1 mm) in independent experiments. At least 14 seeds of each genotype were used in each experiment. Results are presented in Table 1 and Figure 6.

All seeds readily germinated in water with a Final Germination Percent (FGP) of 100%. It has been established that 100% of non-transgenic seeds as well as for all transgenic lines with the only exception for RC12 seeds (93.8%) also germinate in 20% PEG 6000. Therefore, a slightly higher PEG 6000 concentration (25%) was used for further experiments to reveal the differences in the osmotic stress tolerance between the studied lines.

The next experiment was performed under the 25% PEG 6000 treatment with two types of seeds: freshly collected (within a month) and seeds, which were collected and stored for a year before the experiment (aged seeds). First, seeds were germinated in PEG for 21 days and the number of germinated seeds was counted. Then, stress was reduced by diluting PEG with water and, after 4 days, the number of germinated seeds was counted again. This allowed us to identify the percentage of seeds that survived the stress conditions. This approach also ensures that possible stress-avoidance mechanisms, based on temporary growth suppression under stress conditions, are not overlooked [41]. In the experiment with aged seeds, 42.9% of non-transgenic seeds germinated in 25% PEG 6000. Five out of six transgenic lines displayed better germination results under osmotic stress, while the RC12 seed germination rate was slightly lower (31.3%). After diluting PEG with water, 92% of the non-transgenic seeds germinated, showing that most of the seeds remained viable after the stress, while ~8% of the seeds probably died. Again, five out of six transgenic lines displayed a higher germination percent, and all *AtAOS*-overexpressing lines demonstrated 100% germination.

In the experiment with fresh seeds, only 26.7% of non-transgenic seeds germinated in 25% PEG 6000. Thus, five out of six transgenic lines displayed better germination results under these conditions, while the RC12 seed germination rate was, again, slightly lower (23.5%). After replacing PEG with water, 86% of the non-transgenic seeds germinated, while 14% of the seeds, most likely, died. Under these conditions, all seeds of transgenic plants have germinated, showing a 100% stress survival rate.

The seed germination rate was recorded in dynamics for fresh seeds of non-transgenic and transgenic lines, which demonstrated the highest percentage of germination under osmotic stress conditions, RA3, RA4, and RC26 (see Table 1) (Figure 6). Observations showed that the difference in germination rate under osmotic stress between non-transgenic seeds and transgenic *AtAOS*-overexpressing seeds becomes noticeable earlier, especially for line RA4, while the difference between the non-transgenic and *AtOPR*-overexpressing RC26 seeds is evident only 8 days after soaking.

#### 3.3.2. Effect of Osmotic Stress on Seedlings Growth

Next, we measured the lengths of roots and shoots of seeds germinated in PEG 6000 solution 7 days after imbibition. The root and coleoptile length of seeds germinated in water was presented in Chapter 3.2.2 (Figure 4). When exposed to osmotic stress, the differences in seedling and root lengths between genotypes became evident despite the considerable variation in the values between individual plants within each genotype especially in transgenic plants (Figure 7 and Appendix A). Although no difference in germination was observed in 20% PEG, RA3, RA4, and RC29 roots and RA4, RA9, and RC26 shoots were significantly longer than those of non-transgenic plants. Similar results were obtained in two other independent experiments. Although the absolute values of seedling growth rate may differ in independent experiments, the tendency for faster growth of transgenic plant seedlings, especially roots, was maintained (Appendix A).

Since a significant portion of seeds did not germinate in 25% PEG, and germinated plants had very short emerged organs, we measured the lengths of the roots and coleoptiles after the stress was removed, i.e., 4 days after diluting PEG with water. The experiment was repeated with freshly collected and aged seeds (Figure 8 and Appendix A). Median length values of transgenic lines were higher than of non-transgenic one, both in fresh and aged seeds. Statistically significant length differences were observed for roots, coleoptiles, or both for RA3, RA4, RC26, and RC29 lines. The most significant length differences were observed in the *AtAOS*-overexpressing lines RA3 and RA4.

## 4. Discussion

Jasmonates are well-known as natural plant growth regulators that play an important role in mediating plant defense against various environmental stresses; hence, proper exogenous application or endogenous production of those substances can enhance crop productivity under unfavorable climatic conditions [7,17,22]. Among cultivated crops, tetraploid emmer wheat has received far less attention from researchers. As was demonstrated previously [37], plants of emmer wheat (cv. Runo) respond more uniformly to mechanical wounding displaying less variations in the hormone levels compared to hexaploid wheat. That makes it a convenient object for studying the jasmonate system, which is known to be induced by mechanical damage of plant tissues.

Although the emmer wheat cv. Runo seemed to be an excellent research object for expressing various heterologous sequences with high capacity for biolistic-mediated transformation [35], the insertion of the *AtAOS* gene into the emmer wheat genome became problematic. The efficiency of cv. Runo transformation using pBAR-GFP.UbiOPR3 construct (*AtOPR3*) was similar to the transformation using reporter genes only [35], as the number of primary transgenic plants was easily generated using a frequency of 13.1% [37]. In contrast, an acceptable number of independent *AtAOS*-positive events was only obtained after the significant increase in the number of explants, while the efficiency of transgenic plants production decreased to 1.7%. Moreover, a significant portion of primary transgenic events with *AtAOS* insertion (three out of nine) showed no expression of the gene of interest, while 84% of transgenic plants successfully carrying the *AtOPR3* gene demonstrated accumulation of *AtOPR3* mRNA in leaf extracts [37]. Both the *AtOPR3* and the *AtAOS* genes were driven under the same strong *Ubi* promoter of maize; it was suggested that transient accumulation of *AtAOS* gene product in emmer wheat cells after biolistic delivery of pBAR-GFP.UbiAOS construct might negatively influence the re-differentiation of cells suppressing morphogenic processes and regeneration of transgenic plants.

One of the main results of the present work is the production and comparative analysis of transgenic emmer wheat plants with the endogenous jasmonate content altered by the overexpression of jasmonate biosynthesis genes, *AtAOS* and *AtOPR3*. The overexpression of both genes in transgenic emmer wheat leads to increased wounding-induced levels of jasmonic acid and jasmonate-isoleucine (Figure 3). Against expectations, the overexpression of *AtAOS*, encoding a chloroplast-localized enzyme, does not result in an increase in the level of the chloroplast-formed OPDA even after leaf wounding. Instead, more JA and JA-Ile accumulate in the tissues in comparison to non-transgenic plants (Figure 3). This suggests an effective conversion of OPDA to downstream products in wounded emmer wheat leaves. Interestingly, there is a tendency for a decrease in the basal level of JA and JA-Ile in transgenic lines in comparison to non-transgenic plants, that show a significant increase in the level of these metabolites in wounded leaves. Similarly, the decreased basal level of JA and JA-Ile was observed in previously generated RC29 and RC26, while the RC29 line displayed the highest expression level of the transferred *AtOPR3* gene [37]. The reduced basal level of jasmonate in intact leaves of some lines with constitutive overexpression of jasmonate biosynthesis genes indicates the presence of the complex regulatory network involving active feedback loops and/or possible gene silencing.

To study the effects of endogenously produced jasmonates on plants under normal condition, we focused on leaf length measurements. Leaf length parameter is easy to characterize due to its high constancy under given growth conditions, whereas other growth parameters were highly variable and difficult to phenotype, especially after the emergence of tillers. The leaf-size parameter is also easy to measure. During the analysis of the leaf length of transgenic lines with increased jasmonate content, it was found that the first and second leaves were shorter compared to those of non-transgenic plants (Figure 5). This is consistent with previously published data that indicated jasmonate-induced inhibition of plant organ growth [16,42,43]. At the same time, the fourth leaf in transgenic plants tends to be longer; also, the tendency towards greater root length was observed for the germinating transgenic seeds (Figure 4 and Figure 5). The opposite changes in the length of the first, second, and fourth leaves in the transgenic lines compared to the non-transgenic control may be the result of altered activity in the jasmonate system at different growth stages and in different organs. There is also a chance that the activity of jasmonate-regulated processes in emmer wheat could be modulated by other factors, including endogenous ones related to plant ontogenesis. It is also difficult to explain the greater root length in germinating transgenic plants in comparison to Runo, since most studies have previously shown a suppressing effect of jasmonates on root growth [44,45,46]. The overall picture may be more complex, since the stimulatory effect of jasmonates on the growth of the sixth seminal root of wheat has already been shown [47].

The effect of osmotic stress on the plants was studied using young seedlings, which tend to be most sensitive to stressors. There is also less variability in wheat plants at early stages of development. We observed an improvement in tolerance to the osmotic stress in transgenic plants overexpressing the jasmonate biosynthesis pathway genes, which was evident both when plants were germinated in 20% PEG 6000 and 25% PEG 6000 (Table 1, Figure 6, Figure 7, Figure 8 and Appendix A). Two types of plant responses were observed using 25% PEG. In one case, seedling growth under the 25% PEG condition was limited to the length less than 1 mm for a long period (21 days), and the growth was only activated after diluting the PEG solution with water. This type of response may be related to so-called stress-avoidance mechanisms [41]. In the experiment using aged seeds, the germination was initiated under stress conditions, the size of germinated roots reached about 5 mm, but then the growth was inhibited until the removing of stress conditions. The observed variance in responses is most likely related to the seed age, as the role of jasmonates in the regulation of seed germination has been shown previously [47], although the influence of other factors, such as the ontogenetic history of seeds and external conditions, cannot be excluded. The protective effect of endogenously produced jasmonates under osmotic stress, expressed as root elongation, was evident despite some differences in the responses of seeds from different stocks. The findings contradict the known effect of jasmonates on root growth suppression, which has been repeatedly shown in Arabidopsis [42,44]. The difference in effect may be due to the differences in types of root system, the tap root system in dicotyledons, and the fibrous root system in monocotyledons.

The effects of jasmonates under stress conditions are diverse; in that case we can speak of a jasmonate-regulated alteration of the cell transcriptional profile and the central metabolism rewiring, that may lead to the induction of a protective state possibly at the expense of growth and development [16]. It seems that inhibition of growth processes does not necessarily accompany the activation of jasmonate-regulated protective mechanisms: we could see the roots and the fourth leaf of jasmonate-accumulating transgenic plants grew equally faster under normal and osmotic stress conditions.

The evidence of the protective functions of endogenously produced jasmonates in wheat under abiotic stress conditions is particularly interesting in the light of our recently demonstrated effects of jasmonates in wheat under biotic stress conditions. Overexpression of the jasmonate biosynthesis gene in hexaploid and tetraploid wheat has been shown to compromise tolerance to *Botrytis cinerea*, whereas in *A. thaliana* jasmonates act as potent protectors against this pathogen [37].

The present work suggests that it is not acceptable to mechanically transfer knowledge of the functioning of the hormone system obtained in one plant species to the rest of the species. From the results of this study, we have been able to provide more knowledge about how the jasmonate system works in wheat, which may be useful in the transition to the practical application of these powerful regulators to control wheat growth and stress tolerance.

## 5. Conclusions

This article shows that endogenously produced jasmonates can affect the growth and stress tolerance of tetraploid wheat plants. Of particular interest is the fact that the effects induced by jasmonates in this monocotyledonous crop are different from those described for dicotyledonous Arabidopsis. The manifestation of jasmonate effects may vary depending on the specific organ in a single plant or at different developmental stages. Our results reveal the diversity of jasmonate-regulated effects and species-specific differences in the jasmonate system.

## Figures and Tables

**Figure 1 biomolecules-13-01775-f001:**
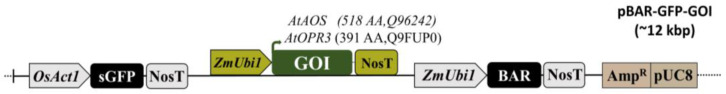
Schematic representation of the pBAR-GFP.UbiAOS expression vector (constructed on the basis of the pUC8 plasmid) used for emmer wheat transformation. *OsAct1*, rice Actin 1 promoter; *ZmUbi1*, maize Ubiquitin 1 promoter; *NosT*, Nopaline Synthase terminator; *sGFP*, modified (S65T) Green Fluorescent Protein gene; *BAR*, BASTA resistance gene (phosphinothricin acetyl transferase); *Amp^R^*, ampicillin resistance gene.

**Figure 2 biomolecules-13-01775-f002:**
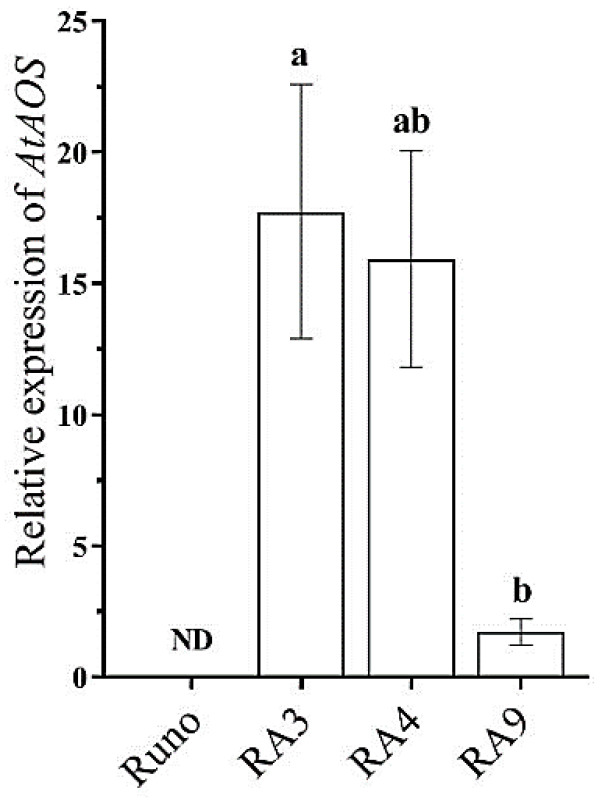
qRT-PCR analyses of the *AtAOS* gene expression in leaf tissues of the transgenic homozygous sublines RA3, RA4, and RA9. The data were normalized to the wheat *TaWIN1* gene. Values are means ± SE of five biological replicates and three technical replicates for each sample. Different letters indicate a significant difference at *p* < 0.05. Accumulation of specific *AtAOS* transcript was not detected (ND) in total RNA extracts of non-transgenic plants of emmer wheat Runo.

**Figure 3 biomolecules-13-01775-f003:**
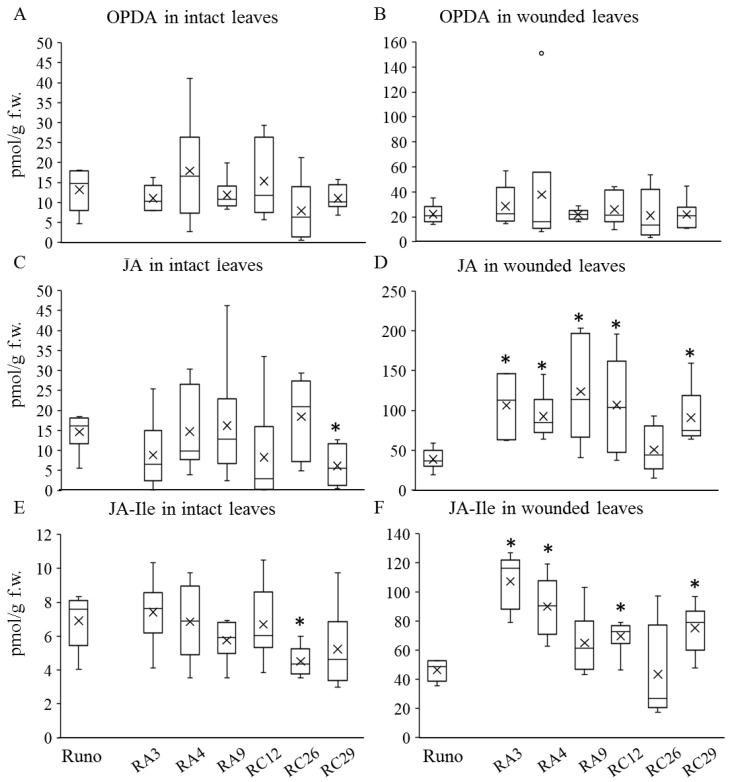
The content of jasmonates ((**A**,**B**) OPDA; (**C**,**D**) JA; and (**E**,**F**) JA-Ile) in intact (**A**,**C**,**E**) and wounded (**B**,**D**,**F**) leaf tissues of non-transgenic emmer wheat (Runo) and transgenic plants with *AtAOS* (RA3, RA4, and RA9) and *AtOPR3* (RC12, RC26, and RC29) gene overexpression. Each box represents data from six biological replicates with whiskers extended to the extreme data points; the midline is the median, the cross is the mean, and the dots are outliers. Stars indicate a statistically significant difference from the non-transgenic control at *p* ≤ 0.05, as assessed using Student’s *t*-test.

**Figure 4 biomolecules-13-01775-f004:**
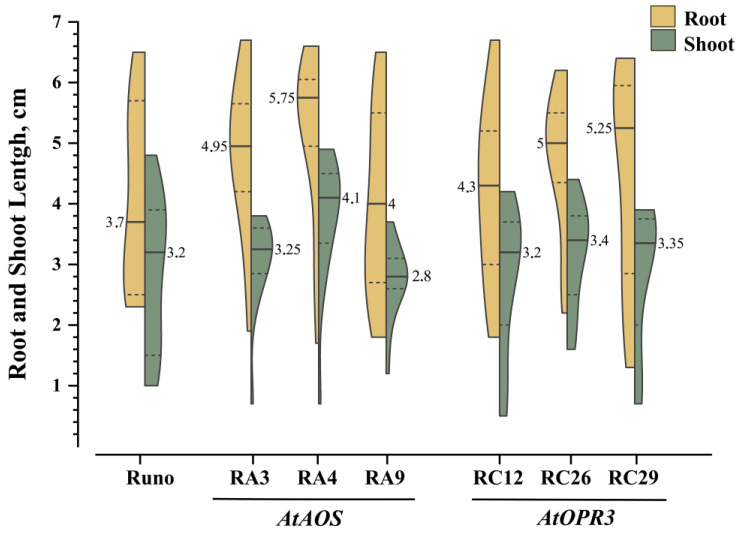
Root and shoot length of transgenic seedlings grown in Petri dishes. Split violin plots display the length of the longest roots (yellow color) and coleoptiles (green color) on the 4th day after the seeds were soaked in water. Each split violin represents data from 17–25 seedlings; the solid horizontal lines within the split violin show the median values and the dotted lines depict the 75th and 25th percentile of the distribution.

**Figure 5 biomolecules-13-01775-f005:**
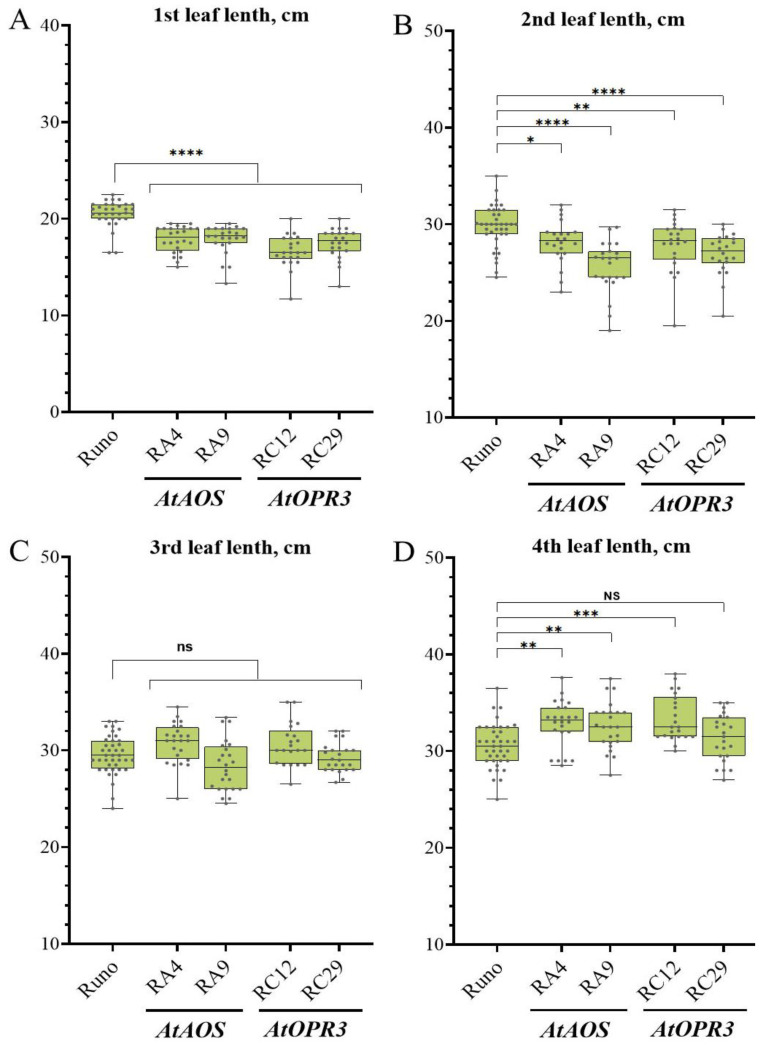
Analysis of leaf length of non-transgenic emmer wheat (Runo) and transgenic plants with *AtAOS* (RA4 and RA9) and *AtOPR3* (RC12 and RC29) gene overexpression. The lengths of 1st (**A**), 2nd (**B**), 3rd (**C**), and 4th (**D**) leaves were measured; values represent averages of 22–24 plants (transgenic lines) or 38 plants (non-transgenic (Runo)). Stars indicate statistically significant differences calculated according Dunnett’s multiple comparison test: (“*”, *p* < 0.05), (“**”, *p* < 0.01), (“***”, *p* < 0.005), and (“****”, *p* < 0.001), (ns, non-significant).

**Figure 6 biomolecules-13-01775-f006:**
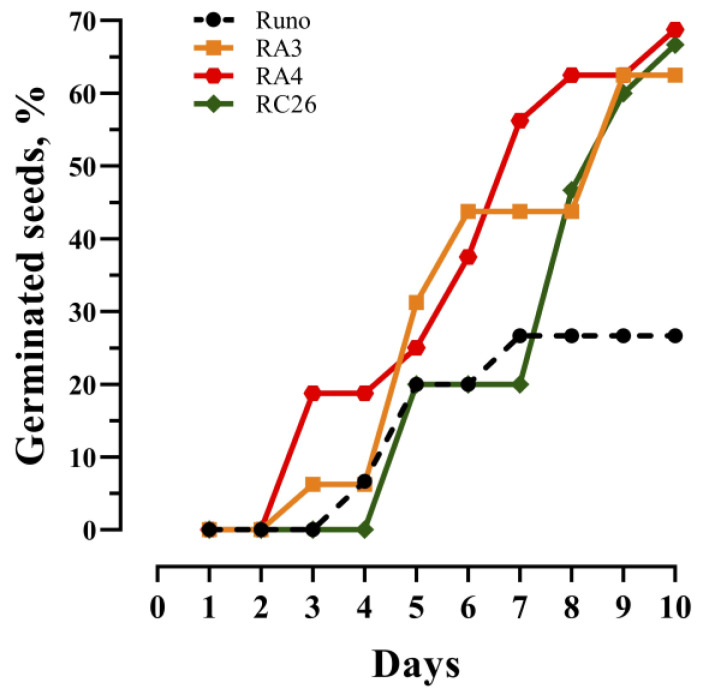
Germination kinetics of emmer wheat seed of non-transgenic (Runo) and transgenic plants overexpressing *AtAOS* (RA3 and RA4) and *AtOPR3* (RC26) under osmotic stress (25% PEG 6000). At least 14 seeds of each genotype were used.

**Figure 7 biomolecules-13-01775-f007:**
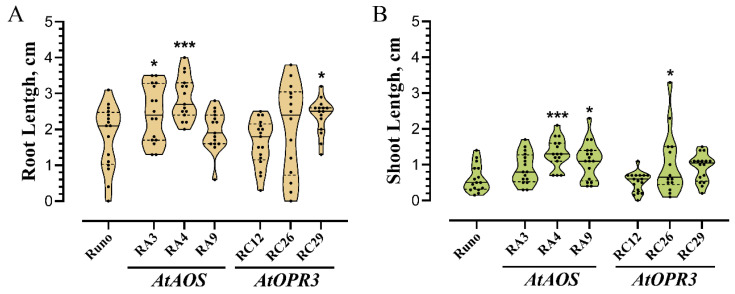
The growth rate of transgenic plant seedlings overexpressing *AtAOS* (RA3, RA4, and RA9) and *AtOPR3* (RC12, RC26, and RC29) and non-transgenic plants (Runo) under osmotic stress conditions. Violin plots display the length of the longest roots (yellow color, (**A**)) and coleoptiles (green color, (**B**)) grown in 20% PEG 6000. Each violin represents data from 14–20 seedlings; the solid horizontal lines within the violin show the median values, the dotted lines depict the 75th and 25th percentile of the distribution, and dots are the values of individual measurements. Stars indicate statistically significant difference between genotypes determined via one-way analysis of variance (ANOVA) followed by a Tukey’s post hoc test (*, *p* ≤ 0.05; ***, *p* ≤ 0.001).

**Figure 8 biomolecules-13-01775-f008:**
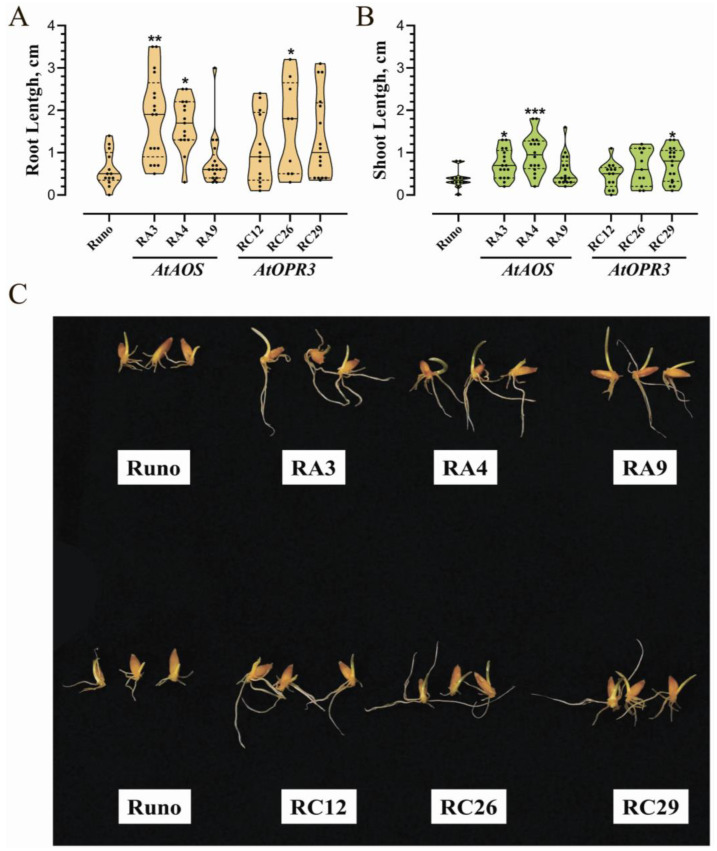
The growth of seedlings germinated from aged seeds of transgenic plants overexpressing *AtAOS* (RA3, RA4, and RA9) and *AtOPR3* (RC12, RC26, and RC29) and non-transgenic plants Runo under osmotic stress conditions. Violin plots display the length of the longest roots (yellow color, (**A**)) and coleoptiles (green color, (**B**)) grown after 21 days of incubation in 25% PEG 6000 followed by the 4 days of incubation after dilution the PEG solution with water. Each violin represents data from 17–25 seedlings; the solid horizontal lines within the violin show the median values, the dotted lines depict the 75th and 25th percentile of the distribution, and dots are the values of individual measurements. Stars indicate statistically significant differences between genotypes determined in one-way analysis of variance (ANOVA) followed by a Tukey’s post hoc test (“*”, *p* ≤ 0.05; “**”, *p* ≤ 0.01; “***”, *p* ≤ 0.001). (**C**) Representative images of seedlings after the stress treatment.

**Table 1 biomolecules-13-01775-t001:** Seed germination under osmotic stress conditions.

Germination, %						
Plant	Water (3–4 Days)	PEG 20% (7 Days)	PEG 25%			
			Aged Seeds		Fresh Seeds	
			PEG (21 Days)	After Adding Water (4 Days)	PEG (21 Days)	After Adding Water (3 Days)
**Runo**	100	100	42.9	92	26.7	86
**RA3**	100	100	58.8	100	62.5	100
**RA4**	100	100	75.0	100	68.8	100
**RA9**	100	100	83.3	100	33.3	100
**RC12**	100	93.8	31.3	87	23.5	100
**RC26**	100	100	43.8	100	66.7	100
**RC29**	100	100	55.6	95	43.8	100

## Data Availability

The presented data are available upon reasonable request from the corresponding author.

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
