# Peer review of "Endogenously Produced Jasmonates Affect Leaf Growth and Improve Osmotic Stress Tolerance in Emmer Wheat"

_biomolecules, 2023, doi:10.3390/biom13121775_

Round 1

Reviewer 1 Report

Comments and Suggestions for Authors

Dear Authors

The Manuscript titled “Endogenously produced jasmonates affect leaf growth and improve osmotic stress tolerance in emmer wheat” focuses on the possibility of improving the stress tolerance of emmer wheat by activating the synthesis of the stress hormone jasmonate by overexpressing two genes of the jasmonate biosynthetic pathway from Arabidopsis thaliana.

The study is well designed, and I appreciate the authors for this work however, I have few questions and suggestions:

1-    What is the significance and application of the current work?

2-    What is the gap in the study?

3-    What are the recommendations based on this study? 

Comments on the Quality of English Language

Minor editing is required, such as spelling and grammar checking.

Author Response

Dear reviewer!

We express our deep gratitude for the analysis of our work and the questions that make us think more about the value of the submitted work. Below are the answers to each question.

  • What is the significance and application of the current work?

The special significance of the presented work lies in the fact that it reveals species-specific features of the jasmonate system and draws attention to possible differences in the manifestation of jasmonate-induced effects in different tissues, organs, at different stages of ontogenesis, even in case when biosynthetic gene expression is controlled by constitutive promoter. These insights are expressed in the discussion and conclusion.

2-    What is the gap in the study?

We have in our hands a convenient tool, a collection of engineered transgenic wheat plants, that will provide answers to many fundamental and practical questions. At the moment, the work has been focused only on the description of phenomenology. In the future, this trend should change, and the molecular mechanisms responsible for phenotype changes should be characterised. Nevertheless, the presented work describing new manifestations of jasmonate effects is complete and interesting.

3-    What are the recommendations based on this study? 

The present work suggests that it is not acceptable to mechanically transfer knowledge of the functioning of the hormone system obtained in one plant species to the rest of the species. From the results of this study, we have been able to provide more knowledge about how jasmonate system operates in wheat, which may be useful for the transition to the practical application of these powerful regulators to control wheat growth and stress tolerance. This information is added to the article (end of the Discussion section.

Sincerely,

Tatyana Savchenko

Reviewer 2 Report

Comments and Suggestions for Authors

The ALLENE OXIDE SYNTHASE (AtAOS) and OXOPHYTODIENOATE REDUCTASE 3 (AtOPR3) are the important enzyme in jasmonate biosynthetic pathway, which can enhance plant stress tolerance by promoting jasmonate synthesis . In this study, authors investigated stress tolerance of emmer wheat by activating the synthesis of the stress hormone jasmonate by overexpressing AtAOS and AtOPR3 genes. The results indicated that the content of jasmonic acid and jasmonate-isoleucine in leaves was increased under wound-induced and the plant tolerance for osmotic stress was also increased. Although the data are interesting and are presented in a logical order, I think that several points need to be addressed. Please see below my detailed comments and suggestions.

1) In this research, the author chose the AtAOS and AtOPR3 genes from Arabidopsis thaliana to heterologously overexpress into Emmer wheat, but why did they not select AOS and OPR3 genes from Emmer wheat for overexpression experiments ?

2) Line 47, line 55 and line 96, the description ofas reviewed in [4-6], as reviewed in [11] and bombardment as described [16]has some problem, please modify the writing style here.

3) The introduction lacks some descriptions about the relationship between methyl jasmonate and stress resistance, please add this section.

4) This section of line 62-73 should be concise and clear, and should add this section in the first paragraph. please rewrite the introduction section.

5) Line 79, extracted replaces isolated.

6) Line 80, gene specific replaces gene-specific.

7) Line 112 and Line 162, wheat plants should be written wheat.

8) Line 112, the writing is wheat; Line 122, the writing is wheat cv. Runo; Line 129, the writing is emmer wheat. Please confirm the specific format and keep the entire text consistent. In addition, please check and correct other inconsistencies in writing in this manuscript.

9) Line 186, the writing of we used in further studies should be incorrect?

10) In figure 2, the error bar is significantly too long, even exceeding the length of the bar graph, indicating significant variability and poor repeatability in the experimental data. There, please repeat the experimental results and provide the original data. In addition, please check the error analysis of other experimental results and unify the experimental and analytical standards.

11) Line 231, t should be written in italic?

12) Line 273, 30.58 cm replaces 30.58cm.

13) Line 267-269, why is there no significant differences in the length of the 3 rd leaves were found?

14) Line 374-377, adding a literature after this sentence.

15) Line 512, 520, 522, 551, 554, 557, Please check and modify the format.

16) Line 548, 551, 567, 573, The Latin name of plants should be italicized.

Please check and modify all reference format. 

Comments on the Quality of English Language

The writing of the manuscript should be improved. Indeed, numerous English errors are present in the manuscript, so I suggest the authors to use an editing service. 

Author Response

Dear reviewer!

We express our deep gratitude for the very insightful analyses of our work and valuable suggestions. Below are responses to each of the comments.

  • In this research, the author chose the AtAOSand AtOPR3 genes from Arabidopsis thaliana to heterologously overexpress into Emmer wheat, but why did they not select AOS and OPR3 genes from Emmer wheat for overexpression experiments ?

Thank you for raising this important issue. We expressed these particular genes for a reason. One of the main reasons has to do with the fact that Arabidopsis genes are well studied and it has been shown that these particular genes/proteins are involved in jasmonate biosynthesis. A number of enzymes of oxylipin biosynthesis pathways, including AOS, are CYP74 proteins, while enzymes of different branches of oxylipin biosynthesis pathway, such as allene oxide synthase and hydroperoxide lyase, may differ by only a few amino acids. Using the same substrate, they form different products, and their product specificity must be established experimentally. The OPR protein family in wheat is also quite large, and some of these enzymes are not relevant to jasmonate biosynthesis. The use of well-studied Arabidopsis genes in this work allows us to avoid possible errors in gene selection. An additional difficulty in cloning the genes from emmer wheat cv. Runo is that its genome has not been sequenced.

A sentence about the selection of the Arabidopsis AOS gene is added to the beginning of paragraph 3.1. Generation of Transgenic Emmer Wheat Plants Overexpressing AtAOS (line 232)

2) Line 47, line 55 and line 96, the description of“as reviewed in [4-6], as reviewed in [11] and bombardment as described [16]”has some problem, please modify the writing style here.

The sentences are corrected.

  • The introduction lacks some descriptions about the relationship between methyl jasmonate and stress resistance, please add this section.

The introduction section was supplemented with information on the role of jasmonates (including methyl jasmonate) in the regulation of plant stress tolerance. Some of the information was added anew, some of it was transferred from the discussion chapter. (lines 81-96)

  • This section of line 62-73 should be concise and clear, and should add this section in the first paragraph. please rewrite the introduction section.

We have rewritten this section to make it shorter and clearer, and moved it to the beginning of the Introduction chapter. (line 40)

Also corrected technical mistakes:

5) Line 79, extracted replaces isolated.

6) Line 80, gene specific replaces gene-specific.

7) Line 112 and Line 162, wheat plants should be written wheat.

8) Line 112, the writing is wheat; Line 122, the writing is wheat cv. Runo; Line 129, the writing is emmer wheat. Please confirm the specific format and keep the entire text consistent. In addition, please check and correct other inconsistencies in writing in this manuscript.

9) Line 186, the writing of “we used in further studies” should be incorrect?

11) Line 231, t should be written in italic?

10) In figure 2, the error bar is significantly too long, even exceeding the length of the bar graph, indicating significant variability and poor repeatability in the experimental data. There, please repeat the experimental results and provide the original data. In addition, please check the error analysis of other experimental results and unify the experimental and analytical standards.

We have repeated experiment using five biological replicas using previously collected and frozen tissues. New graph is presented in new version of the manuscript.

12) Line 273, 30.58 cm replaces 30.58cm.

We found that “the International System of Units (SI) recommends inserting a space between a number and a unit of measurement units and between units in compound units”. So we have left it unchanged, but we are willing to change it if it needs to be done to suit the requirements of the journal.

13) Line 267-269, why is there no significant differences in the length of the 3 rd leaves were found?

Indeed, no statistically significant difference was found in the third leaf length, while the second and fourth leaves show the opposite trend. We are sure that this is not an artefact, but so far, we do not know how to explain the phenomenon. Most likely, the degree of silencing of jasmonate responses changes with the plant aging. This question requires further research. The information about it is in the Discussion section (lines 538-543).

14) Line 374-377, adding a literature after this sentence.

The references are added (line 479)

We have corrected all reference format as requested in comments 15 and 16:

15) Line 512, 520, 522, 551, 554, 557, Please check and modify the format.

16) Line 548, 551, 567, 573, The Latin name of plants should be italicized.

Please check and modify all reference format. 

Sincerely,

Tatyana Savchenko

Reviewer 3 Report

Comments and Suggestions for Authors

The article is based on the testing of newly created emmer wheat transformants with a gene for jasmonic acid synthesis (AOS overexpressors of the enzyme from arabidopsis). The transformants are compared with previously prepared overexpressors of another gene of the same pathway (OPR3). The authors compared shoot (and individual leaves) and root growth and germination rates. Germination under osmotic stress conditions was also tested. The data looks interesting, showing a new pattern of organ growth caused by jasmonic acid. Nevertheless, these results should be better supported by other methods (at least the other methods used in the article). The biggest weaknesses are in the lack of gene expression under different conditions. I do not know if the same generation was used for the gene expression. However, the expression should be done for the wounding experiment as well as for the growth and germination experiments. It is essential because the potential of gene silencing can be really high in the case of this kind of overexpressors. Moreover, the wounding experiment really needs this kind of correlation.

It is also a pity that the hormonal analysis was done only for the third leaf because the results indicate different jasmonic acid effects (accumulation?) in different leaves. As the authors said, the unchanged levels in control plants might be caused by active downregulation of JA synthesis. It could be nice to measure some metabolites of genes connected with JA metabolism. But I understand that it is hard to measure these selected genes (like ILLs or at least JAR) in wheat and I do not know if the sequences are known. And I also understand that the measurement of JA metabolites is difficult.

I have some other notes:

Abstract - l. 25 - the wounding is confusing in the text of abstract and this sentence should be better explained to the readers who have not yet read the paper

Introduction - l. 41 - references 2 and 3 are about different topics

l. 62, 227 - correct 2 parentheses

l. 68 - correct vitamine PP to B3; some citations could be good in this paragraph

Methods - add the information about the number of samples (missing in Fig. 3) and the number of independent repetitions of the experiments - independent experiments are necessary in the case of stress conditions (wounding, osmotic stress)

Add the description of wounding experiment.

l. 95 - add a short description of the tissue which was used for the transformation

l. 128 - add a short description of the type of analysis used for phytohormone detection (HPLC)

l. 136, 149 - correct M to mol

l. 168 - missing pUC8 abbreviation in the legend

Results

Fig. 3 - add a short description of the variants (like in Fig. 5)

l. 224 - it is not clear why you suggest the tendency for a decrease in JA levels. Please, explain it better. The note about the potential silencing should be added to the discussion as well as the gene expression data.

Table 1 - add numbers of seeds used for the test

Fig. 6 - it is not clear which stress was used

l. 334 - what do you mean that you counted shoot and root length after 7-8 days?

l. 335 - correct chapter 3.2.2

l. 339 - correct seedlings to shoots

Discussion

The growth of plants in the germination experiment might be affected also by volatile compounds (methylated JA). Did you have the plants on different dishes or did you put randomly different genotypes on the same dish?

The discussion about the root growth should also include the potential difference between monocots and dicots as is mentioned in Conclusion.

l. 406 - correct the OPDA whole name to the abbreviation

l. 409 - add a reference

l. 416 - add the info about potential silencing

l. 436 - correct germinal to seminal

l. 462 - the reference 34 is about rice, not peanut

Author Response

Dear reviewer!

We express our deep gratitude for the very insightful analyses of our work and valuable suggestions. Thank you for the identification of errors and for the opportunity to correct them. We've done everything we can so far. We have significantly revised the article based on the reviewers' comments. Below are responses to each of the comments.

Nevertheless, these results should be better supported by other methods (at least the other methods used in the article). The biggest weaknesses are in the lack of gene expression under different conditions. I do not know if the same generation was used for the gene expression. However, the expression should be done for the wounding experiment as well as for the growth and germination experiments. It is essential because the potential of gene silencing can be really high in the case of this kind of overexpressors. Moreover, the wounding experiment really needs this kind of correlation.

It is also a pity that the hormonal analysis was done only for the third leaf because the results indicate different jasmonic acid effects (accumulation?) in different leaves. As the authors said, the unchanged levels in control plants might be caused by active downregulation of JA synthesis. It could be nice to measure some metabolites of genes connected with JA metabolism. But I understand that it is hard to measure these selected genes (like ILLs or at least JAR) in wheat and I do not know if the sequences are known. And I also understand that the measurement of JA metabolites is difficult.

Thank you for this important comment. We fully agree with the remarks about the importance of analyzing the expression of endogenous genes due to the possibility of silencing jasmonate responses. In addition to silencing of the AOS gene (most likely several AOS genes), the constitutive expression of the transgene can lead to the silencing of the jasmonate system at other stages of hormone biosynthesis, as well as changes in the catabolism of active jasmonates and at the level of signalling. We plan to address this challenge in a comprehensive manner, most likely using an RNA-Seq approach, as the Runo cultivar used has not been sequenced yet. At this time, it is not possible to perform the proposed gene expression analysis studies as well as more detailed jasmonate analyses. We were only able to repeat the gene expression analysis of the transferred gene in a larger number of biological replicates using previously collected and frozen tissues. Please, consider this work without additional gene expression analyses.

Abstract - l. 25 - the wounding is confusing in the text of abstract and this sentence should be better explained to the readers who have not yet read the paper

The explanation about mechanical damage to leaves has been added to the abstract. The sentence is now written as follows: “Analyses of jasmonates in intact and mechanically wounded leaves of non-transgenic and transgenic plants showed that the overexpression of each of the two genes resulted in an increased wounding-induced levels of jasmonic acid and jasmonate-isoleucine in leaves”. Line 25

Introduction - l. 41 - references 2 and 3 are about different topics

We are very sorry for the error and thank you for finding it. These should be references to the websites that were automatically replaced with random links. All references have now been saved as text, and the website links have been corrected (references 5 and 6 in the modified version of the article).

  1. 62, 227 - correct 2 parentheses

We have included a more complete species name using square brackets within the name. We are willing to change the writing according to the journal's requirements (lines 43-44).

  1. 68 - correct vitamine PP to B3; some citations could be good in this paragraph

The sentence is modified to make it more concise (vitamins and micronutrients are not listed, just mentioned that grains are rich in vitamins and micronutrients), and references are added (lines 47-49).

Methods - add the information about the number of samples (missing in Fig. 3) and the number of independent repetitions of the experiments - independent experiments are necessary in the case of stress conditions (wounding, osmotic stress)

We completely agree with the reviewer, independent repetition of any experiments on plant stress response is critical for the result quality. This basic requirement was, of cause, addressed in our work. Thus, all wounding experiments were set up in six independent repetitions. For this, prior to the experiments the plants were randomized and wounded in different times (keeping however, the constant time interval between wounding and harvesting). The corresponding changes are done in the text (lines 173-175).

Relevant information on the replication of the osmotic stress experiments has also been added to the end of paragraph “2.5. Osmotic stress tolerance test” and to the Results section (line 223-229), to the beginning of paragraph “3.3.1.        Effect of osmotic stress on seed germination” (lines 380-383), and figure legends.

Add the description of wounding experiment.

The description of wounding experiment is added to the Material and Methods section (2.3. Phytohormone analysis) (lines 173-178).

  1. 95 - add a short description of the tissue which was used for the transformation

10-15 day old scutellum–derived morphogenic calli were used. The information is added (line 137).

  1. 128 - add a short description of the type of analysis used for phytohormone detection (HPLC)

In modified version of the manuscript, we provide a short description of the method with a reference to a recent paper where all instrument settings are listed. The provided information then is absolutely sufficient for the adequate reproduction of our study (lines 178-195).

  1. 136, 149 - correct M to mol

Thank you. We apologize for the mistake. It is corrected.

  1. 168 - missing pUC8 abbreviation in the legend

The information “constructed on the basis of the pUC8 plasmid” is added to the legend.

Results

Fig. 3 - add a short description of the variants (like in Fig. 5)

The information is added (lines 313-315).

  1. 224 - it is not clear why you suggest the tendency for a decrease in JA levels. Please, explain it better. The note about the potential silencing should be added to the discussion as well as the gene expression data.

The required information is added: “The median JA and JA-Ile values in 5 out of 6 transgenic lines are lower than in the non-transgenic control, with this the difference being statistically significant in two transgenic lines (RC29 – in JA, RC26 – in JA-Ile)” (lines 308-310).

Table 1 - add numbers of seeds used for the test

Relevant information on the replication of the osmotic stress experiments has also been added to the end of paragraph “2.5. Osmotic stress tolerance test” and to the Results section (line 223-229), to the beginning of paragraph “3.3.1.        Effect of osmotic stress on seed germination” (lines 380-383), and figure legends.

Fig. 6 - it is not clear which stress was used

It was 25% PEG 6000. The information is added to the figure legends.

  1. 334 - what do you mean that you counted shoot and root length after 7-8 days?

We measured the length of roots and shoots of seeds germinated on PEG 6000 solution 7 days after imbibition. This information is more clearly presented in the corrected version of the manuscript (lines 431-432).

  1. 335 - correct chapter 3.2.2

Thank you. The error is corrected.

  1. 339 - correct seedlings to shoots

Thank you. The mistake is corrected.

Discussion

The growth of plants in the germination experiment might be affected also by volatile compounds (methylated JA). Did you have the plants on different dishes or did you put randomly different genotypes on the same dish?

We are aware of the possible effects of volatile compounds, so seeds of only one genotype were placed in each dish. Relevant information has been added to Part 2.5. Osmotic stress tolerance test (line 212).

The discussion about the root growth should also include the potential difference between monocots and dicots as is mentioned in Conclusion.

Thank you for the suggestion. This information has been added to Discussion section: “The protective effect of endogenously produced jasmonates under osmotic stress, expressed as root elongation, was evident despite some differences in the responses of seeds from different stocks. The findings contradict the known effect of jasmonates on root growth suppression, which has been repeatedly shown on Arabidopsis [49,50]. The difference in effect may be due to the differences in root systems, the tap root system in dicotyledons and the fibrous root system in monocotyledons” (lines 566-572).

  1. 406 - correct the OPDA whole name to the abbreviation

It is corrected

  1. 409 - add a reference

This conclusion follows from the data presented in this paper and not from the literature.

  1. 416 - add the info about potential silencing

The information is added (line 525).

  1. 436 - correct germinal to seminal

Thank you. It is corrected.

  1. 462 - the reference 34 is about rice, not peanut

Thank you! It is corrected (line 89).

Thank you again for the valuable comments.

Sincerely,

Tatyana Savchenko

Round 2

Reviewer 2 Report

Comments and Suggestions for Authors

This manuscript has been carefully revised by the author and have reached the level of publication in the magazine.

Reviewer 3 Report

Comments and Suggestions for Authors

I agree with the corrections.

Small corrections:

l. 358 - you probably mean "leaf blade"

l. 660 - correct "length of non-transgenic"